# OpenReview forum: "Reward Auditor: Inference on Reward Modeling Suitability in Real-World Perturbed Scenarios"
_ICML.cc/2026/Conference — ICML 2026 regular_

### Official Review · Reviewer_q8Ef · 2026-02-16

**Soundness:** 3
**Presentation:** 3
**Significance:** 3
**Originality:** 3
**Overall Recommendation:** 5
**Confidence:** 3

**Summary:**

The paper introduces Reward Auditor which is a hypothesis testing framework to evaluate how suitable reward models are under "real-world" perturbations. The way suitability is defined is conditional reliability. That is, reward models ought to maintain the preference-perception confidence which is user-defined when inputs or outputs are perturbed in realistic ways. They use a paired hypothesis testing framework to quantify the p-values and effect sizes and apply multiple hypothesis testing control. The authors perform case studies across rewardbench subsets. They find that reward models are vulnerable to small perturbations. They also find a strong negatiave correlation between measured suitability risk and downstream policy robustness in perturbed training settings.

**Compliance With Llm Reviewing Policy:**

Affirmed.

**Final Justification:**

The responses have resolved my concerns, thanks for the authors.  I'm increasing my score from a 3->5 since all the questions are addressed well. I hope these are included in the final version too.

**Key Questions For Authors:**

- The formal hypothesis H1 is stated in terms of first-order stochastic dominance (FSD), but the implemented test primarily evaluates mean shifts via a paired t-type statistic under permutation. There is a mismatch between the stated property (FSD) and the tested one (mean difference), and no dedicated dominance test (e.g., one-sided signed-rank, KS-type, or quantile-based dominance) is used.
- Would be good to understand how you decide effect size thresholds or what is a good effect size in this case.
- Is there any huiman validation on stylized response perturbations not introducing subtle drifts in the meaning of the sentences?
- Comparability across RM families is nontrivial. For DPO-based implicit rewards, assuming πref=1 when unavailable changes the reward shape; for generative RMs, verdict-token probabilities can be template- and tokenization-sensitive. More calibration discussion or cross-family normalization/ablation would help.
- You state H1 in terms of stochastic dominance but implement a mean-shift test. Why not use a dominance-aligned test (e.g., one-sided signed-rank, one-sided KS, or quantile-based tests)? If mean shift is the intended target, can you restate the hypotheses accordingly?
- What specific value(s) of the margin m did you use when declaring "insufficient suitability"? If not used, can you report decision outcomes for a set of m values and provide a sensitivity analysis?

**Limitations:**

- Recent auditing work using hypothesis testing for LM robustness (e.g., Rauba et al., 2025) and classifier-suitability frameworks (Pouget et al., 2025) are cited, but the paper overstates being "first" to transform evaluation into inference; a sharper positioning against these lines and other distributional testing (e.g., dominance or OT-based comparisons) is warranted. Could you e.g. add a related work table comparing this?

**Strengths And Weaknesses:**

- Clear and practical framework
- The proposed multiplicity-control mechanism seems apt and suitable for the multi-scenario auditing setting
- Broad evaluation framework (26 diverse reward models), evaluated over 10 perturbations
- Other mini things, e.g. running normality checks to justify non-parametric inference, which is a very nice addition and often lacking in machine learning papers
- the RLHF case study is well received
- Clearly presented, auditing pipeline is described very clearly
- Generally, I believe this framework can inform model selection
- By now it's standard to use LLM-as-a-judge, so the GPT-5 being used is fine with me.

Overall, the methodological core—paired permutation testing with effect sizes and multiplicity control—is sound and well-motivated, and the empirical sweep across many RMs and perturbations yields actionable insights, particularly about over-sensitivity to stylistic variations and language changes.

---

> ### Author Rebuttal · Authors · 2026-03-30
>
> Dear Reviewer q8Ef,
>
> Thank you for your valuable feedback. Our responses are as follows:
> ## Question1&5
> Regarding your **concern about the form of $H_1$ FSD** in Question1 & 5, we appreciate you pointing this out. FSD in the paper was mainly intended to convey the intuition of directional degradation, but from a strict statistical perspective, **this formulation is indeed too strong**.
>
> Based on the definition of suitability, our formulation defines the paired degradation variable $M-M'$, and assesses significance via permutation test. Strictly, this tests whether perturbations cause **systematic directional average degradation**. Therefore, in the revised version, we will adjust the hypothesis formulation:
> $H_0: \mathbb{E}[M-M']=0, H_1: \mathbb{E}[M-M']>0$.
> This more accurately reflects Reward Auditor's role as a risk auditing tool: identifying whether perturbations cause significant and practically meaningful confidence degradation, rather than directly conducting a dedicated test for the complete distribution dominance relationship.
> ## Question2&6
> Regarding your **concern about $\hat{e}$ and $m$** in Questions2 & 6. We will supplement discussion in the revised version:
>
> First, While $\alpha$ controls **statistical significance**, $m$ is a threshold for the effect size $\hat e$, used to characterize **the practically tolerable degradation margin**. Therefore, unlike $\alpha$, $m$ does not have a universal default value. It should be set by users based on specific task scenarios and tolerance levels.
>
> Second, in downstream alignment experiments, we employed a conservative empirical rule to set $m$: for a perturbation, we set $m$ by referring to **the maximum effect size among models whose degradation is not yet statistically significant**. Intuitively, this is equivalent to treating the maximum non-significant degradation as the upper bound of still tolerable fluctuations for that scenario（e.g. suppose the suitability risk reports of 6 RMs are $0.20^{\*\*}, 0.15^{**}, 0.10^{*}, \underline{0.005}, 0.003, 0.001$, then we set $m$ to 0.005.）.
>
> ## Question3
> Regarding your **concern about the semantic-preserving of the perturbations**, when applying perturbations, we **strictly follow the methods from related work, which have already shown these perturbations are semantic-preserving**. Specifically: ST defined in the literature as meaning-preserving word replacement[1]; LE is controlled expansion-style paraphrase, aiming to change linguistic attributes while keeping meaning[2]; SP involves style rewriting, which reorganizes expression without altering core information[3]; LC/SLC use translation-based transformations, aiming to express the same content in another language[4].
>
> Furthermore, **the assumption that such perturbations should remain semantically consistent is typically a default premise in existing RM hacking and robustness research**[4][5].
>
> [1] Representing word meaning in context via lexical substitutes
>
> [2] Quality Controlled Paraphrase Generation
>
> [3] SC2: Towards Enhancing Content Preservation and Style Control in Text Style Transfer
>
> [4] M-RewardBench
>
> [5] RM Bench
> ## Question4
> Regarding your **concern about cross-family RM evaluation**, we want to emphasize that **unified evaluation across different types of RMs is already common practice in recent RM evaluations** (e.g.RewardBench, RM-Bench, RMB). We also want to emphasize that Reward Auditor uses a standardized paired effect size, which measures the magnitude of the average degradation relative to the model's own variability scale. This forms a **dimensionless metric** that is fully suitable for cross-family comparison.
>
> We also conducted consistency experiments on chat subset, comparing suitability risk rankings from two DPO reward parameterizations and two generative RM prompt settings. The results shown in the cross-setting Spearman correlation matrix (0–1), reveal overall high correlations, indicating that the main conclusions are highly robust to DPO parameterization and generative RM template choices.
> |Setting|S1|S2|S3|S4|
> |-|-|-|-|-|
> |**S1** Our setting|1.000||||
> |**S2** Ref-aware DPO|0.9993 |1.000|||
> |**S1** Criteria-aware prompt|1.000|1.000|1.000||
> |**S2** Minimal prompt|1.000|1.000|1.000|1.000|
> ## Limitations
> Regarding your **concern about the presentation of contribution**,
> actually, we do not claim to be the first to transform general model evaluation into statistical inference. As stated in the paper  (line 047-049, right column): "...first framework to transform **RM evaluation** into...".
> We summarize the differences between these works in the table below, emphasizing that we are **the first to introduce statistical inference in RM evaluation**.
> |Work|Object|Audit|
> |--|--|--|
> |Reward Auditor|RM|Preference confidence degradation|
> |Pouget et al.|Classifier|Accuracy degradation during deployment|
> |Rauba et al.|LLM|If perturbations change output distribution|

---

> > ### Author Rebuttal · Reviewer_q8Ef · 2026-04-03
> >
> > This has resolved my concerns, thanks for the authors.  I'm increasing my score from a 3->5 since all the questions are addressed well. I hope these are included in the final version too.

---

### Official Review · Reviewer_XPxX · 2026-02-28

**Soundness:** 2
**Presentation:** 3
**Significance:** 2
**Originality:** 3
**Overall Recommendation:** 4
**Confidence:** 3

**Summary:**

The paper proposes Reward Auditor, a hypothesis testing framework designed to evaluate the conditional reliability, defined as suitability, of reward models under various input perturbations. By employing paired permutation tests, the authors aim to identify systematic vulnerabilities in reward models that static accuracy metrics might overlook. The study applies this framework to multiple reward models using the RM Bench dataset and reports that suitability risk correlates with the performance of downstream policy models.

**Compliance With Llm Reviewing Policy:**

Affirmed.

**Final Justification:**

The paper tackles an important problem by shifting reward model evaluation from static accuracy to conditional reliability under perturbations. The proposed framework is conceptually interesting, and the empirical results—especially the stronger correlation with downstream robustness—support its potential significance.

My initial concerns were about the definition of suitability, the realism of perturbations, and the necessity of the statistical framework. The rebuttal has partially addressed these points, particularly by clarifying the null hypothesis and demonstrating advantages over simpler metrics. This improves my confidence in the method.

Some concerns remain regarding the gap to real-world distributions and the distinction between desirable sensitivity and fragility. Overall, the rebuttal has positively influenced my assessment, leading me to slightly increase my score, though I still see room for further refinement.

**Key Questions For Authors:**

## Questions

1. How does your framework distinguish between a 'vulnerable' model (one that is fragile to irrelevant noise) and a 'sensitive' model (one that correctly detects a drop in quality)?

2. Could similar diagnostic insights be achieved using simpler, more interpretable metrics (e.g., average margin drop or accuracy degradation)? What specific 'latent vulnerabilities' does the hypothesis testing framework uncover that a direct analysis of the confidence distribution shifts would miss?

**Limitations:**

The paper addresses a valid problem in reward model evaluation but suffers from methodological choices that limit its practical utility and reproducibility. The definition of suitability requires refinement to distinguish between fragility and correct sensitivity to quality changes.

**Strengths And Weaknesses:**

## Strengths
The paper is well-written with gradual justification, and the motivation is clearly demonstrated. The focus from simple preference accuracy is shifted to the dynamic reliability of reward models under perturbation, which is interesting and promising. The paper address a critical issue regarding the safety and robustness of alignment systems, as latent vulnerabilities in reward models can lead to undesirable behaviors in large language models. The rigorous statistical formulation, utilizing nonparametric paired tests, provides a formalized mechanism for quantifying confidence degradation.

## Weaknesses
1. The manuscript relies on strong assumptions on perturbations. The definition of suitability implies that a robust reward model should maintain consistent confidence across original and perturbed inputs. However, stylized perturbations, such as length extension or language conversion, can inherently alter the quality or readability of a response. If a perturbation legitimately degrades the quality of a response, a high quality reward model should arguably reflect this with lower confidence or a lower score. The proposed framework potentially penalizes this desirable sensitivity, conflating necessary quality awareness with model fragility.

2. The reliance on synthetic and programmatic perturbations to represent real world scenarios is a limitation. While the authors distinguish between controlled and stylized perturbations, these automated augmentations did not fully capture the complexity and nuance of natural distribution shifts found in authentic user interactions. Consequently, the claim of auditing suitability in real world scenarios appears overstated.

3. The statistical machinery introduced is quite complex. The necessity of the elaborate hypothesis testing framework, involving paired permutation tests and group aware multiplicity control, is not fully justified against simpler and more interpretable metrics. Direct measurement of the average degradation in reward margin or accuracy drops could potentially offer similar diagnostic insights without the high overhead of the proposed statistical apparatus.

---

> ### Author Rebuttal · Authors · 2026-03-25
>
> Dear Reviewer XPxX,
>
> Thank you for your valuable feedback. Our responses are as follows:
> ## Weakness1
> Regarding your **concern that our hypothesis is too strong**, first we want to emphasize that this null hypothesis $H_0:M \overset{d}{=} M'$ **does not require "maintaining consistent confidence" but rather posits that the two sets of confidence come from the same distribution**. That is: there is no systematic difference between the original confidence distribution and the perturbed distribution. This is the **standard exchangeability null hypothesis in hypothesis testing** （For example, in a longitudinal study of plant growth, researchers measure the same group of plants over a one-year interval to test for differences in their growth distributions. The null hypothesis does not assume zero growth; rather, it posits that observed changes stem from random fluctuations rather than systematic bias. The goal is to identify whether such changes are statistically significant.）.
> The null hypothesis is only rejected when the degradation is statistically significant $(\hat p < \alpha)$, indicating that the degradation is objective.
>
> Second, we would like to clarify that Reward Auditor does not evaluate simple score drops for individual responses; instead, it assesses the stability of preference perception confidence. Therefore, our framework audits the **stability of preference judgments, not the fluctuations in the absolute scores of individual responses**.
> ## Weakness2
> Regarding your **concern about our perturbation setting and "real-world scenarios" statement**, we will refine in the revised version: we audit **"suitability under realistically motivated perturbation scenarios for actual deployment"**. This aligns with the perspective in WildBench: real in-the-wild user distributions are far more complex than any fixed synthetic test suite[8].
>
> However, we also want to clarify that these perturbations are not arbitrarily constructed synthetic noise, but are designed around realistic variation axes that repeatedly appear in existing work as shown in following table and are relevant to real-world deployment.
> |Perturbation|Description|
> |--|--|
> |EF/PH|Differences in formatting and punctuation habits in user inputs[3][5]|
> |CN|Typos in user inputs[2]|
> |IU/IW|Common irrelevant additional information contamination in user interactions[1]|
> |ST|Inconsistent lexical expression across different LLM responses[1][2]|
> |LE|Same LLM answers output at different lengths[6]|
> |SP|LLM responses with Markdown formatting[3][4]|
> |LC/SLC|LLM responses appearing in different languages from the prompt /and with Markdown-formatted presentation[7]|
>
> [1] Beyond Accuracy: Behavioral Testing of NLP Models with CheckList
>
> [2] TextFlint
>
> [3] How I Learned to Start Worrying About Prompt Formatting
>
> [4] Does Prompt Formatting Have Any Impact on LLM Performance?
>
> [5] A Large-Scale Comparison of Prompt Robustness Methods for LLMs
>
> [6] A Long Way to Go: Investigating Length Correlations in RLHF
>
> [7] M-RewardBench
>
> [8] WildBench
> ## Weakness3 & Question2
> Regarding your **concern about the Reward Auditor metrics**, first, the computational overhead of Reward Auditor's metrics is almost negligible compared to the inference cost of the RM itself (as summarized in following table).
>
> Second, the paper has directly compared the simple metric (accuracy drop) with Reward Auditor's metric Suitability risk report,  and found that Reward Auditor's metrics have a stronger correlation with the robustness of policy models in downstream alignment training (please refer to **Section5.2, SectionB.4**) . We also supplemented experiments regarding the average confidence degradation and summarized in following table, this indicates that compared to simple metrics, the suitability estimated by Reward Auditor better demonstrates: **"RMs with high suitability can train policy models with high robustness"**.
> |Metric|End-to-end evaluation (s)|RM inference (s)|Permutation test (s)|Group-aware BH (s)|Spearman correlation coefficient (absolute value)|
> |-|-|-|-|-|-|
> |Accuracy drop|72.77|72.32| \ | \ |0.261|
> |Average confidence degradation|72.94|72.60| \ | \ |0.594|
> |**Suitability risk report**|73.62|72.51|1.09|0.02| **0.881**|
> ## Question1
> Regarding your **concern about labeling RM as "vulnerable" or "sensitive,"** Reward Auditor addresses the question: when given a perturbation scenarios, does the preference confidence distribution undergo **systematic** and **practically** significant degradation:
>
> (1) The p-value reflects whether this degradation is statistically significant, not merely random fluctuation (i.e., **determining if the RM is fragile to irrelevant noise**);
>
> (2) The effect size reflects whether the magnitude of this degradation exceeds the tolerable boundary (i.e., **quantifying the drop in quality**).
>
> Only by integrating these two aspects does the framework support the conclusion of insufficient suitability.

---

> > ### Author Rebuttal · Reviewer_XPxX · 2026-04-03
> >
> > Thanks for your response and I will increase my rating accordingly.

---

### Official Review · Reviewer_rMRH · 2026-03-12

**Soundness:** 4
**Presentation:** 3
**Significance:** 3
**Originality:** 4
**Overall Recommendation:** 5
**Confidence:** 3

**Summary:**

Reward models (RMs) are central to RLHF-based alignment, yet their standard evaluation relies on preference accuracy over clean datasets, a setting that may not reflect how they behave when inputs are slightly perturbed in realistic deployment scenarios. This paper introduces Reward Auditor, a framework for statistically auditing whether a reward model's preference judgments remain stable under such perturbations.
The core contribution is the notion of Suitability: a model is considered suitable if its confidence distribution over preference pairs does not degrade meaningfully when inputs are perturbed. This is operationalized through a paired permutation test with a tolerable degradation margin, combined with a group-aware Benjamini–Hochberg correction to handle multiple perturbation scenarios. The authors instantiate this framework across 10 perturbation types covering both prompt-side noise and stylistic response transformations, and apply it to audit 26 reward models across RM Bench domains.
Beyond the auditing framework itself, the paper presents evidence that the aggregated suitability risk scores correlate with downstream policy robustness in an RLHF training experiment, suggesting the metric has practical predictive value beyond its diagnostic use.

**Compliance With Llm Reviewing Policy:**

Affirmed.

**Final Justification:**

Having gone through the authors' rebuttal carefully, I feel my concerns have been well addressed, and I've updated my scores to reflect that, bringing my overall recommendation up from 3 to 5. My initial reservations were mostly around the soundness of the methodology and how clearly the contributions were positioned relative to prior work. The rebuttal tackled both of these directly and convincingly. The technical clarifications resolved the ambiguities I had pointed out, and I've raised my soundness score accordingly.
On originality, the rebuttal did a much better job of drawing out what makes this work distinct, and I found that persuasive.
The presentation improvements I'm hoping for are largely about reflecting the clarity the authors showed in their responses. They clearly understand the material well, and the final version of the paper should feel that way too.
Overall, this was a constructive rebuttal that genuinely changed my assessment. I think this will be a strong addition to the conference.

**Key Questions For Authors:**

1. Why were exactly 10 perturbation scenarios chosen? Would results change with fewer or more perturbations?
2. How should practitioners choose the degradation margin m in practice?
3. What tool or model was used to generate stylized perturbations?
4. Were any checks performed to verify semantic equivalence of perturbations?

**Limitations:**

The evaluation relies partly on proprietary models (GPT-5) and on stylized perturbations whose generation process is not fully described. These factors may affect reproducibility and interpretation of the results.

**Strengths And Weaknesses:**

Strengths and Weaknesses
Soundness
The paper's core statistical machinery is principled and appropriate. Framing reward model robustness as a hypothesis testing problem, using paired permutation tests to compare confidence distributions before and after perturbation, is a well-motivated choice given the non-parametric nature of the data. The application of Benjamini–Hochberg correction to control false discovery rate across multiple perturbation scenarios reflects genuine statistical care.
That said, several soundness concerns limit confidence in the conclusions. The degradation margin m is central to the suitability decision rule, yet the paper provides insufficient guidance on how it should be set and does not analyze how sensitive model rankings are to different choices of m. This is a meaningful gap because the binary classification of models as suitable or unsuitable depends directly on this threshold. Additionally, the stylized perturbation generation process is underspecified: without knowing what model or constraints were used, and without any semantic equivalence validation, it is unclear whether confidence drops after perturbation reflect genuine RM vulnerability or simply correct responses to semantically altered inputs. The downstream evaluation relying exclusively on GPT-5 as a judge also raises reproducibility concerns, since the model is proprietary and may change over time.
Presentation
The paper is generally well organized and the high-level narrative is easy to follow. The progression from motivation to framework definition to empirical audit to downstream validation is logical. The introduction of Suitability as a formal concept is clearly stated, and the use of RM Bench as the evaluation substrate is well contextualized.
However, several presentation gaps reduce clarity. The rationale for selecting exactly 10 perturbation scenarios is never explained, which makes it difficult to assess whether the experimental design is principled or ad hoc. Appendix D, which presumably describes stylized perturbation generation, is insufficiently detailed for reproducibility. The paper would also benefit from a clearer discussion of the directional nature of the hypothesis test and why improvements under perturbation are explicitly excluded from the framework, as this asymmetry has conceptual implications that deserve acknowledgment rather than implicit treatment.
Significance
The problem this paper addresses is genuinely important. As reward models are increasingly deployed in production RLHF pipelines, understanding their behavior under realistic input variations is a critical practical concern that static preference accuracy cannot capture. The framework's ability to surface vulnerability patterns across 26 models and multiple domains provides comparative insights that the community currently lacks.
The downstream experiment linking suitability risk to policy robustness is the most significant empirical contribution, as it grounds the auditing metric in something practitioners care about directly. If this correlation holds reliably across settings, Reward Auditor could become a useful pre-deployment checklist tool. The significance is somewhat limited by the reproducibility concerns around GPT-5 evaluation and the underspecified perturbation methodology, both of which would need to be resolved before the framework could be broadly adopted with confidence.
Originality
The paper's originality lies in its reframing of reward model evaluation as a statistical auditing problem rather than a benchmark accuracy problem. This is a meaningful conceptual shift. Prior work on reward model evaluation has largely treated robustness informally or through aggregate accuracy metrics; the introduction of Suitability as a formally defined, inferentially grounded property is a genuine contribution.
The combination of paired permutation testing, effect size thresholding, and multiplicity-aware correction applied specifically to reward model confidence distributions is novel even if each individual component is well established. The paper also breaks new ground in connecting auditing outcomes to downstream RLHF policy behavior, a link that has not been systematically studied. The originality would be strengthened by a clearer positioning against related work on adversarial robustness of LLMs and preference learning, ensuring readers understand precisely what distinguishes this auditing perspective from prior robustness analyses.

---

> ### Author Rebuttal · Authors · 2026-03-27
>
> Dear Reviewer rMRH,
>
> Thank you for your valuable feedback. Our responses are as follows:
> ## Question1
> Regarding your **concern about selecting these 10 perturbations**, we selected them because they can compactly cover the classic perturbations that have been repeatedly discussed in existing work and are closely related to real-world deployment (refer to response to Reviewer XPxX Weakness2).
>
> As the number of scenarios and null hypotheses increases, the group-aware BH procedure automatically increases the strength of evidence required to determine significant degradation, thereby better controlling multiplicity. Therefore, increasing perturbation scenarios doesn't exaggerate RM vulnerability; instead, it **makes the determination of RM suitability degradation significance ($\mathbb{I}^*(\hat{p},\alpha)$) more conservative**.
> ## Question2
> Regarding your **concern about margin $m$**, we will provide a more detailed discussion in revised versions.
> First, $m$ plays a different role from the significance level $\alpha$. While $\alpha$ controls statistical significance, **$m$ is a threshold for the effect size $\hat{e}$, used to characterize the practically tolerable degradation margin**. Therefore, unlike $\alpha$, $m$ does not have a universal default value. It should be set by users based on specific scenarios and tolerance levels. For example, when auditing a set of RMs, users can choose a more strict or lenient $m$ according to how conservative a standard they wish to adopt.
>
> Second, in downstream alignment experiments, we employed a conservative empirical rule to set $m$: for a specific perturbation scenario, we set $m$ by referring to the maximum effect size among models whose degradation is not yet statistically significant. Intuitively, this is equivalent to treating the **maximum non-significant degradation** as the upper bound of still tolerable fluctuations for that scenario (e.g. suppose the suitability risk reports of 6 RMs are $0.20^{\*\*}, 0.15^{**}, 0.10^{*},\underline{0.005},0.003, 0.001$, then we set $m$ to 0.005.).
> ## Question3
> Regarding your **concern about the tools used to generate perturbations**, we present them in the table below and will add detailed discussion (including tools' parameters, prompt templates, etc.) in the revised version.
> |Perturbation|Method|
> |-|-|
> |EF|**CheckList[1]** based rules for format rewriting|
> |PH|**CheckList** based rules to simulate punctuation habit changes|
> |IU|**CheckList** based rules to append irrelevant strings|
> |IW|**CheckList** based rules to append random URLs|
> |CN|**TextFlint[9]** for character replacement and swapping|
> |ST|**TextFlint** for synonym replacement|
> |LE|**GPT-5** for controlled expansion|
> |SP|**GPT-5** for structured rewriting|
> |LC| **GPT-5** for translation|
> |SLC|**GPT-5** for translation and structured rewriting|
>
> ## Question4
> Regarding your **concern about semantic preservation of perturbations**, first, we strictly followed the methods from related work when constructing the perturbations, and these methods have been confirmed to be semantic-preserving in the literature. For example, CheckList treats invariance tests as semantic-preserving perturbations[1]; similarly, lexical substitution[2], controlled paraphrasing[3], style rewriting[4], and translation-based transformations[5] are all defined with semantic preservation as their objective.
>
> Moreover, this **semantic-preservation assumption is already a default premise in existing RM hacking and robustness research**[5][6], where semantically consistent perturbations are routinely used to test whether RMs exhibit unstable preference behavior under non-semantic changes.
>
> ## Limitations
> Regarding your **concern about using GPT-5 as the judge model**,
> first, as Reviewer q8Ef mentioned, **LLM-as-a-judge is the mainstream paradigm in current open-ended preference evaluation** [7][8].
> Second, our downstream evaluation conclusions do not depend on a single judge model. Here we supplement a consistency experiment in the Tab. below.
> |Judge pair|Raw agreement(%)|Cohen's κ|
> |--|--|--|
> |GPT-5 vs GPT-5.1|99.8|0.996|
> |GPT-5 vs GPT-4o|99.5|0.988|
> |GPT-5 vs Gemini-3.1-pro|98.6|0.965|
> |GPT-5 vs Qwen3-70B|98.5|0.961|
>
>  GPT-5 shows high consistency with its different versions (GPT-5.1, GPT-4o) and other closed and open-source models (Gemini-3.1-pro, Qwen3-70B), with raw agreement reaching 98.5%–99.8% and Cohen’s κ reaching 0.961–0.996. This result indicates that **our evaluation conclusions are highly robust to the choice of judge**.
>
>  ## References
> [1] Beyond Accuracy: Behavioral Testing of NLP Models with CheckList
>
> [2] Representing word meaning in context via lexical substitutes
>
> [3] Quality Controlled Paraphrase Generation
>
> [4] SC2: Towards Enhancing Content Preservation and Style Control in Text Style Transfer
>
> [5] M-RewardBench
>
> [6] RM Bench
>
> [7] Judging LLM-as-a-Judge with MT-Bench and Chatbot Arena
>
> [8] JudgeLM: Fine-tuned Large Language Models are Scalable Judges
>
> [9] TextFlint

---

> > ### Author Rebuttal · Reviewer_rMRH · 2026-03-31
> >
> > Thank you for the thorough and constructive review. The authors' responses address the core concerns well, particularly the judge consistency table showing κ above 0.96 across GPT-4o, Gemini, and Qwen3-70B, which largely resolves the reproducibility concern around GPT-5 evaluation. The clarification on m as an effect size threshold rather than a significance parameter is also helpful and should be incorporated into the main text.
> > The remaining gaps around perturbation methodology detail and sensitivity analysis on m are straightforwardly addressable in revision, and the core contributions of the statistical auditing framework, suitability metric, and downstream correlation remain solid. Given this, the paper could reasonably be reconsidered as a weak accept if the authors follow through on the clarifications promised in the rebuttal.

---

> > > ### Author Response · Authors · 2026-04-03
> > >
> > > Dear Reviewer rMRH,
> > >
> > > Thank you very much for your follow-up feedback. We are glad that our rebuttal helped address your main concerns and sincerely appreciate your recognition of the core contributions of this paper.
> > >
> > > Below, we briefly respond to your concerns about the **details of the perturbation methods** and the **further analysis of $m$**, which we could not fully present in the rebuttal due to space limits. We **commit to clearly adding these details and the clarifications made in the rebuttal, in the revised version**.
> > > ## Details of the Perturbation Methods
> > > |Perturbation|Tool & Implementation|Parameter Settings|
> > > |--|--|--|
> > > |EF| CheckList-based rule rewrite |only one emphasis format is applied to each sample; the emphasis is limited to the main instruction or the first core clause |
> > > |PH| CheckList-based punctuation rule |modify 1–3 punctuation neighborhoods per sample; character-level edit ratio ≤ 5% |
> > > |IU| CheckList-based template insertion |append only one irrelevant username at the end of the sentence; username length 8–12 characters; fixed format `@` + alphanumeric string |
> > > |IW| CheckList-based template insertion |append only one irrelevant URL at the end of the sentence; fixed format `https://example.com/<slug>`; `slug` length 6–12 |
> > > |CN| TextFlint `Keyboard` / `SpellingError` / `light delete-swap` |total character edit ratio ≤ 2%; operations are limited to keyboard-neighbor replacement / adjacent swap / single-character deletion|
> > > |ST| TextFlint `SwapSynWordNet` |only content words are replaced; named entities, numbers, and proper nouns are not replaced; skip if no high-confidence synonym is available|
> > > |LE| GPT-5, plain-text expansion|single generation; deterministic setting (`temperature=0`); target length about 1.3–1.8× of the original text; Markdown is not allowed; prompt template follows RM-Bench |
> > > |SP| GPT-5, Markdown rewrite | single generation; deterministic setting; length controlled to 1.3–1.8× of the original text; headings / lists / bold are allowed; no facts may be added or removed; the prompt template follows the Markdown axis in RM-Bench, i.e., corresponding to its `“remove the Markdown formatting … without altering the content,”` and we use the reverse rewrite: `“add Markdown formatting without altering content”` |
> > > |LC| GPT-5, translation only | single generation; deterministic setting; one target language is fixed in each experiment; Markdown is not allowed; preserve the original factual content, stance, and preference semantics; prompt design follows the style-controlled rewrite in M-RewardBench; principle: change only the language, not the content|
> > > |SLC| GPT-5, translation + Markdown |single generation; deterministic setting; one target language is fixed following M-Reward Bench; Markdown is allowed; length controlled to 1.3–1.8× of the original text; no new facts may be added; the template follows both the Length and Markdown control axes in RM-Bench|
> > >
> > > ## Further Analysis of the Degradation Margin $m$
> > > As we explained in the rebuttal, $m$ is not a significance parameter like $\alpha$. Instead, it is a **practical significance threshold** applied to the effect size $\hat e$. Therefore, changing $m$ does not alter the underlying statistical evidence produced by the audit itself: once the audit is completed, both $\hat p$ and $\hat e$ are fixed. The role of $m$ is only to affect the final binary decision through $(\hat p<\alpha)\wedge(\hat e>m)$.
> > >
> > > To make this point more concrete, we conduct a sensitivity analysis using the EF perturbation in Table 1 of the paper as an example. Following the heuristic rule proposed in our rebuttal, we define the baseline threshold as
> > >
> > > $m_{\text{base}}=\max(\hat e^i:\hat p^i\ge\alpha)$,
> > >
> > > that is, the **largest effect size among models that have not yet reached statistical significance**. We then scale this baseline threshold by different factors and examine whether the proportion of models classified as suitable changes substantially under different thresholds.
> > > |Threshold $m$|Suitable Ratio|
> > > |---|---|
> > > |$0.5m_{\text{base}}$|92.3%|
> > > |$0.75m_{\text{base}}$|92.3%|
> > > |$m_{\text{base}}$|92.3%|
> > > |$1.25m_{\text{base}}$|92.3%|
> > > |$1.5m_{\text{base}}$|92.3%|
> > > |$1.75m_{\text{base}}$|92.3%|
> > > |$1.9m_{\text{base}}$|96.2%|
> > > |$2m_{\text{base}}$|100.0%|
> > > This table shows that the conclusion is **piecewise stable rather than arbitrary** with respect to $m$. In particular, within a fairly broad and reasonable range around $m_{\text{base}}$, the suitable classification under the EF scenario remains unchanged. This indicates that the main conclusion does not fluctuate under small adjustments of the threshold. In the revised version, we will include a complete sensitivity analysis of $m$ using Table 1 as an example.
> > > ## We sincerely hope that our clarifications and planned revisions have adequately addressed your concerns. If you feel they do, we would be very grateful if this could be reflected in your updated assessment.

---

### Official Review · Reviewer_pg9N · 2026-03-12

**Soundness:** 3
**Presentation:** 4
**Significance:** 3
**Originality:** 4
**Overall Recommendation:** 5
**Confidence:** 4

**Summary:**

This paper introduces Reward Auditor, a framework to assess the suitability of reward models under perturbed conditions, with particular attention to the alignment application domain. The authors propose measuring the difference in the reward model’s confidence between preference judgments for chosen and rejected responses, using a hypothesis testing framework. They show that reward models are often not robust given stylised perturbations such as rephrasings and word substitutions with their synonyms. They also find that this type of suitability risk is a more solid predictor of downstream alignment performance than traditional accuracy metrics.

**Compliance With Llm Reviewing Policy:**

Affirmed.

**Final Justification:**

I confirm my original recommendation for this paper (5: Accept). The authors have answered all my questions and have reinforced my positive opinion of the work.

**Key Questions For Authors:**

- Would Reward Auditor still be useful for RL training runs where the data is expected to be clean? The suitability audit seems to reveal whether reward models rely on superficial features rather than semantic understanding, which seems valuable regardless of data cleanliness.

- It seems the same framework could also be applied to inference-time LLM-as-a-judge evaluators to verify their robustness and reliance on surface level features. What is the authors’ take on this?

- Similarly, it seems that the framework could potentially be used for any reward model, beyond just alignment, and including online learning, perhaps by taking advantage of the multiple completions per prompt that already happen in GRPO/RLOO. I would be curious to hear the authors’ opinion on a such a potential extension.

**Limitations:**

I would strongly suggest adding a limitations/future work section (or lengthening the conclusion to integrate it) to discuss the potential scope extensions mentioned above. This would not only make the paper stronger but also aid the community by pointing other practitioners toward interesting novel applications of the method.

**Strengths And Weaknesses:**

Strengths:

- Soundness: The authors present a comprehensive evaluation of 26 reward models and five domains. The methodology is statistically rigorous.

- Presentation: The paper is well-structured, with clear language throughout and high-quality, clear figures.

- Significance: The topic is timely and interesting. The authors provide an array of findings that are of interest to the community and are practically actionable, for example that stylised perturbations are more damaging than controlled perturbations, and in particular that reward models are especially vulnerable to synonym swaps and general language changes, and also that suitability risk has strong Spearman correlation with downstream performance, but accuracy has only weak correlation.

- Originality: The idea seems quite novel. I am not aware of another work that frames the same question in statistical inference terms like the authors did here.

Weaknesses:

- Soundness: Perhaps I have missed this, but I think the paper does not explain well how perturbations are applied/controlled. It does not appear that the authors use an established attack framework like TextFooler or similar. How are the perturbations applied and how is perturbation intensity chosen? This is particularly important given that the authors provide a comparison between perturbation types, but without knowing the respective intensities the results could contain bias. I would appreciate if the authors could explain this in their response and also add more detail to the manuscript.

- Significance: The scope is slightly narrow (i.e., a similar strategy could be applied to reward modelling in other domains, and to LLM-as-a-judge evaluation). I do understand the need to narrow down the scope for an 8-page paper but I would still suggest that the authors include the potential other applications of their work in a Limitations/Future work section (which is currently missing).

---

> ### Author Rebuttal · Authors · 2026-03-29
>
> Dear Reviewer pg9N,
>
> Thank you for your recognition of our work and your valuable feedback! Our responses are as follows:
> ## Weakness1
> Regarding your **concern about the tools used to generate perturbations**, we present them in the table below and will add detailed discussion (including tools' parameters, prompt templates, etc.) in the revised version.
> |Perturbation|Method|
> |-|-|
> |EF|Using **CheckList[1]** based rules for format rewriting|
> |PH|Using **CheckList** based rules to simulate punctuation habit changes|
> |IU|Using **CheckList** based rules to append irrelevant strings|
> |IW|Using **CheckList** based rules to append random URLs|
> |CN|Using **TextFlint[2]** for character replacement and swapping|
> |ST|Using **TextFlint** for synonym replacement|
> |LE|Using **GPT-5** for controlled expansion|
> |SP|Using **GPT-5** for structured rewriting|
> |LC|Using **GPT-5** for translation|
> |SLC|Using **GPT-5** for translation and structured rewriting|
>
> [1] Beyond Accuracy: Behavioral Testing of NLP Models with CheckList
>
> [2] TextFlint
> ## Weakness2 & Question2
> Regarding your **concern about the scope of the current manuscript**, it is indeed intentionally focused on RM in preference alignment. The reason first is that the object audited by Reward Auditor is the degradation of **preference-confidence distribution**, which is built on **continuous reward scores** and derived from reward gaps in the form of Bradley–Terry. In other words, this framework is most naturally applicable to systems that can output **continuous scalar rewards/judgments**. It is precisely for this reason that we have focused this paper on preference-based RM in RLHF.
>
> In a broader sense, LLM-as-a-judge can actually be seen as a special case of generative RMs, albeit one that operates through generative judgments. However, in the current mainstream usage paradigm for preference data, LLM-as-a-judge typically performs **pairwise comparisons** on two candidate answers for the same prompt, rather than producing stable continuous scalar scores. Therefore, such judge settings do not naturally satisfy the formal premise of continuous reward to preference confidence that Reward Auditor relies on.
>
> Nevertheless, as long as the target system can produce **comparable continuous scalar scores**, this audit framework could potentially be extended to LLM-as-a-judge paradigms. A particularly well-suited emerging direction is **rubrics-as-rewards**: in such settings, judges no longer only output discrete win/loss outcomes, but instead provide continuous or quasi-continuous scalar judgments based on scoring rubrics. This makes it easier to form dense reward signals and more naturally adapts to the audit object of Reward Auditor. We will supplement this point in the revised version and explicitly include it in the **Future Work** section.
>
> [1] Rubrics as rewards: Reinforcement learning beyond verifiable domains
>
> [2] Reinforcement learning with rubric anchors
>
> ## Question1
> Regarding your **concern about the value of Reward Auditor in the scenario where RL training data is clean**, we believe it remains valuable. Detecting issues in downstream RL caused by explicit perturbations or dirty data is merely an extended functionality of Reward Auditor. In essence, **Reward Auditor serves as a pre-deployment stress test or a model screening tool**. An RM may perform normally on clean training but still rely on fragile formatting, length, structural, or linguistic cues—issues that often only manifest under natural variations in real-world deployment. The value of Reward Auditor lies in identifying such hidden suitability risks in advance, either before training or deployment.
>
> Compared to controlled prompt-side perturbations, response-level stylized perturbations more directly test whether the RM, when evaluating answer quality, relies on surface expression forms rather than more robust semantic grounding. These perturbations primarily alter the vocabulary, length, structure, and linguistic expression of responses while preserving their core semantics and preference relationships. Therefore, models ranked higher under the stylized perturbations in Table 1 of the paper indicate that in dialogue tasks, they can maintain more stable preference judgments under these response-level stylistic variations.
> ## Question3
> Regarding your **concern about the value of Reward Auditor in other RL scenarios**, we believe it is a valuable direction. For methods like GRPO, multiple completions are naturally generated under the same prompt, which indeed provides a natural candidate structure for Reward Auditor-style auditing.
>
> Of course, as we responded in Weakness2 & Question2, this extension is predicated on the target system being able to provide comparable **dense reward signals** to support the current preference confidence-based formalism. For settings that rely solely on discrete rankings or binary win/loss signals, additional methodological adjustments would be needed.

---

> > ### Author Rebuttal · Reviewer_pg9N · 2026-03-31
> >
> > Thank you for the response and for clarifying what tools were used to generate the perturbation. Could you maybe spend a couple more words regarding perturbation intensity? You find that stylised perturbations are more damaging, which is an interesting finding, however how do you quantify the intensity of the stylised perturbations, and ensure that the intensity level of different types of perturbations is comparable? Thanks

---

> > > ### Author Response · Authors · 2026-04-03
> > >
> > > Dear Reviewer pg9N,
> > >
> > > We are very pleased to have the opportunity for further discussion with you. Regarding your **concerns about perturbation intensity and the comparability of evaluation across perturbation families**, we would like to clarify the following.
> > >
> > > Our core view is that, for heterogeneous perturbation families, there is no universally accepted raw intensity scale that can strictly calibrate all perturbation types to an identical perturbation strength in advance. Synonymous substitution, structural rewriting, length expansion, and translation operate on different variation axes. Many related robustness works themselves focus on multiple real-world scenarios or multiple types of perturbation scenarios, and none can first define a unified original intensity unit across families [1][2].
> > >
> > > Therefore, to achieve reasonable cross-perturbation family evaluation in our task, what truly needs to be unified and controlled is not the so-called original perturbation intensity, but rather a **standardized evaluation under a unified statistical inference protocol**.
> > >
> > > Specifically, the suitability risk report $r_S \triangleq \hat e \wedge \mathbb{I}^{*}(\hat p,\alpha)$ reported by Reward Auditor consists of two parts:
> > >
> > > (1) The standardized effect size $\hat e$ of paired samples;
> > >
> > > (2) The significance marker $\mathbb{I}^{*}(p,\alpha)$ obtained through the same paired test process across all perturbation types.
> > >
> > > Where $\hat e \triangleq \frac{\overline{\Delta M}}{\mathrm{std}(\Delta M)}$, $\overline{\Delta M}$ is the sample mean of the paired difference distribution, and $\mathrm{std}(\Delta M)$ is its sample standard deviation. From this, it can be seen that $\hat e$ is a **dimensionless standardized metric** (the numerator and denominator have the same dimension), so $\hat e$ itself does not carry original units. It normalizes the average degradation magnitude by the fluctuation scale of paired differences within the perturbation scenario, making it more suitable for cross-scenario comparison than the original preference confidence drop.
> > >
> > > At the same time, $\mathbb{I}^{*}(p,\alpha)$ does not measure the degradation magnitude itself, but rather determines whether the degradation has **systematic evidence** that is statistically identifiable under a unified paired test framework. In the paper, it is defined as a step indicator function that returns hierarchical significance markers based on $\hat p$ and the threshold set $\alpha$.
> > >
> > > Therefore, our cross-perturbation scenario comparison is based on two types of standardized results under a unified audit protocol: $\hat e$ provides a comparable characterization of degradation magnitude, and $\mathbb{I}^{\*}(p,\alpha)$ provides a unified inference judgment on whether degradation holds. Based on this, our experimental conclusion is not that stylized perturbations are stronger on some unified original intensity scale. More precisely, our conclusion is: under the current controlled perturbation construction, stylized perturbations induce **larger standardized degradation magnitudes** ($\hat{e}$) in the RM's preference confidence and more frequently exhibit **systematic statistically significant degradation** ($\mathbb{I}^{*}(p,\alpha)$).
> > >
> > > [1] Beyond Accuracy: Behavioral Testing of NLP Models with CheckList
> > >
> > > [2] Models in the Wild: On Corruption Robustness of Neural NLP Systems

---

### Decision · Program_Chairs · 2026-04-30

**Decision:**

Accept (regular)

**Comment:**

This paper reframes reward model evaluation from static accuracy to statistical inference on conditional reliability (Suitability) under realistic perturbations, using paired permutation tests with standardized effect sizes and group-aware BH multiplicity control across 26 RMs and 10 perturbation types. All four reviewers converged on Accept (5) after a rebuttal that resolved the main concerns. Lingering items for the camera-ready: (i) semantic preservation of GPT-5-generated stylistic perturbations relies on prior-work conventions rather than human validation, and a spot-check would strengthen the causal interpretation; and (ii) key rebuttal clarifications should be integrated into the main text alongside an explicit Limitations section. The paper merits acceptance for its novel conceptual reframing, rigorous methodology, broad empirical coverage, and compelling downstream validation.